# Organisational and neuromodulatory underpinnings of structural-functional connectivity decoupling in patients with Parkinson's disease

Angeliki Zarkali [1✉], Peter McColgan[2], Louise-Ann Leyland[1], Andrew J. Lees[3], Geraint Rees [4,5] & Rimona S. Weil[1,5,6]

Parkinson's dementia is characterised by changes in perception and thought, and preceded by visual dysfunction, making this a useful surrogate for dementia risk. Structural and functional connectivity changes are seen in humans with Parkinson's disease, but the organisational principles are not known. We used resting-state fMRI and diffusion-weighted imaging to examine changes in structural-functional connectivity coupling in patients with Parkinson's disease, and those at risk of dementia. We identified two organisational gradients to structural-functional connectivity decoupling: anterior-to-posterior and unimodal-to-transmodal, with stronger structural-functional connectivity coupling in anterior, unimodal areas and weakened towards posterior, transmodal regions. Next, we related spatial patterns of decoupling to expression of neurotransmitter receptors. We found that dopaminergic and serotonergic transmission relates to decoupling in Parkinson's overall, but instead, serotonergic, cholinergic and noradrenergic transmission relates to decoupling in patients with visual dysfunction. Our findings provide a framework to explain the specific disorders of consciousness in Parkinson's dementia, and the neurotransmitter systems that underlie these.

[1] Dementia Research Centre, University College London, 8-11 Queen Square, London WC1N 3AR, UK. [2] Huntington's Disease Centre, University College London, Russell Square House, London WC1B 5EH, UK. [3] Reta Lila Weston Institute of Neurological Studies, 1 Wakefield Street, London WC1N 1PJ, UK. [4] Institute of Cognitive Neuroscience, University College London, 17-19 Queen Square, London WC1N 3AR, UK. [5] Wellcome Centre for Human Neuroimaging, University College London, 12 Queen Square, London WC1N 3AR, UK. [6] Movement Disorders Consortium, University College London, London WC1N 3BG, UK. ✉email: a.zarkali@ucl.ac.uk

D ementia associated with Parkinson's disease (PD) is characterised by changes in cognition and perception, including visual hallucinations, delusions and fluctuations in attention[1,2]. It is often preceded and accompanied by visual dysfunction[3–5] and linked to hypometabolism in posterior brain regions[6]. High-order visual dysfunction, in particular, is associated with worse cognition at 1-year follow up[7]. Although PD is characterised by Lewy body inclusions, the neural correlates of cognitive impairment in PD and specifically the structural and functional changes remain unclear[8].

Perception and action, whether in health or disease, depends on connections between brain regions. In general, it is assumed that there is a relationship between the strength of a structural connection between two brain areas and the strength of the corresponding functional connection[9]. However, it has recently emerged that this relationship between structural–functional connectivity is not uniform across the healthy human brain but organised with clear hierarchical and cyto-architectural principles[9]. Specifically, there is close structural–functional coupling (SC–FC coupling) in primary sensory (unimodal) cortices, with divergence at the apices of processing hierarchies (transmodal association cortices), in networks such as the default mode network (DMN)[10–12]. One theory for this is that relative decoupling in higher-order areas allows abstract reasoning, protected from the more granular signalling in earlier stages of sensory processing[13]. Changes in SC–FC coupling occur during brain maturation[10] but also in psychiatric[14,15] and neurological disease[16–19], and maybe particularly relevant to cognition: individual differences in coupling reflect differences in cognition[20,21] and higher SC–FC coupling in prefrontal cortex is associated with improved executive function[10]. Therefore, loss of SC–FC coupling might be expected in PD, especially in subtypes linked with higher risk of dementia.

Neuroimaging studies have provided important insights separately into structural and functional connectivity alterations in PD[22–24]. Diffusion-weighted imaging revealed structural alterations in tracts including the corpus callosum and thalamo-cortical connections in PD with cognitive impairment[25–29] and those with visual dysfunction (higher dementia risk)[30]. Resting-state functional MRI (rsfMRI) studies have identified changes in functional connectivity between frontal and visuospatial regions[31,32] and frontal regions and the posterior cingulate[7,32] in PD with cognitive impairment. These studies provide useful insights into the network-level dysfunction contributing to cognitive impairment in PD, however, the question of how structural changes impact on brain function is unresolved. We hypothesised that the relationship between structural–functional coupling across the brain would be systematically modified in PD and that this pattern of decoupling would occur along with one of two hypothesised directions: (1) across the unimodal–transmodal hierarchical gradient of SC–FC decoupling that is seen in health with more transmodal regions becoming even more decoupled in PD[10–12,33]; or (2) along the anterior-to-posterior (A–P) axis with decoupling more prominent in posterior regions. This hypothesis was based on the posterior distribution of metabolic and connectivity changes seen in PD [25,30,34–36].

We used rsfMRI and diffusion-weighted imaging to investigate changes in whole-brain structural connectivity–functional connectivity coupling (SC–FC coupling) in 88 patients with PD (of whom 33 had visual dysfunction and higher dementia risk) and 30 age-matched controls. We found widespread decoupling in PD compared to controls but a more focal pattern affecting the insula in PD with visual dysfunction compared with those with normal visual function. Next, we evaluated the specific pattern of decoupling in PD and found that this occurred across both a unimodal–transmodal and anterior–posterior axes.

Finally, we examined whether changes in SC–FC coupling are related to underlying differences in expression of specific neurotransmitters in an exploratory analysis. Although PD is classically associated with the altered dopaminergic transmission, recent evidence implicates other neurotransmitter systems: cholinergic transmission[37–39] is affected in PD dementia and both reduced occipital GABA levels[40] and altered noradrenergic transmission[41] have been implicated in cognitive impairment in PD. We show that dopamine transmission, although central to motor aspects of PD, may have a less important role in PD dementia, as neurotransmitter systems other than dopamine were correlated with the SC–FC decoupling found in PD with visual dysfunction.

## Results

To characterise how structural–functional connectivity (SC–FC) coupling changes in PD, we quantified the degree to which a brain region's structural connectivity relates to coordinated fluctuations in neural activity between-regions. For each participant, two weighted, undirected connectivity matrices were derived using the same parcellation[42] comprised of 400 cortical brain regions: a structural connectivity matrix derived from diffusion-weighted imaging and a functional connectivity matrix derived from resting-state functional MRI (rsfMRI) data. SC–FC coupling was measured as the Spearman rank correlation between the structural and functional connectivity profiles of each region. An overview of the study methodology is seen in Fig. 1.

A total of 118 participants were included: 88 patients with PD and 30 controls. Patients with PD were further classified according to their performance in two higher-order computer-based visual tasks which have been previously shown to correlate with worsening cognition over time[7]. This resulted in 33 PD low visual performers and 55 PD high visual performers.

MRI quality and pre-processing were visually and quantitatively evaluated. Excluding cases with low-quality structural MRI or high head motion on rsfMRI resulted in the exclusion of 14 subjects from our original cohort, leading to the final sample of 88 PD and 30 controls.

Importantly, the three groups did not significantly differ in scan quality, gender and years in education (Table 1). As in previous work[43,44], performance in visual tasks correlated with cognition but not on low-level vision tests such as visual acuity. Details of neuropsychological performance in Supplementary Table 1. PD low and high visual performers were well-matched in disease duration, severity and levodopa equivalent dose (Table 1).

**Widespread structural–functional connectivity decoupling occurs in PD**. First, we examined how the relationship between structural and functional connectivity changes in PD. All participants showed statistically significant correlations between structural and functional connectivity (correlation coefficient range = 0.28–0.74, all $p_{spin} < 0.001$). Similarly to other studies[10,45], controls showed variation in SC–FC coupling across the cortex, with higher coupling in primary sensory and medial prefrontal cortex and lower coupling in lateral temporal and frontoparietal regions (Fig. 2A). This pattern was preserved in PD; however, SC–FC coupling was globally reduced in PD participants compared to controls (mean 0.484 in PD vs 0.544 in controls, $p = 0.002$) (Fig. 2B, C).

When we examined SC–FC coupling in all nodes across the whole brain, 8 nodes showed significantly reduced coupling in PD compared to controls (adjusting for age and gender, FDR-corrected over 400 nodes $q < 0.05$). The nodes showing SC–FC decoupling in PD had a posterior distribution: bilateral superior

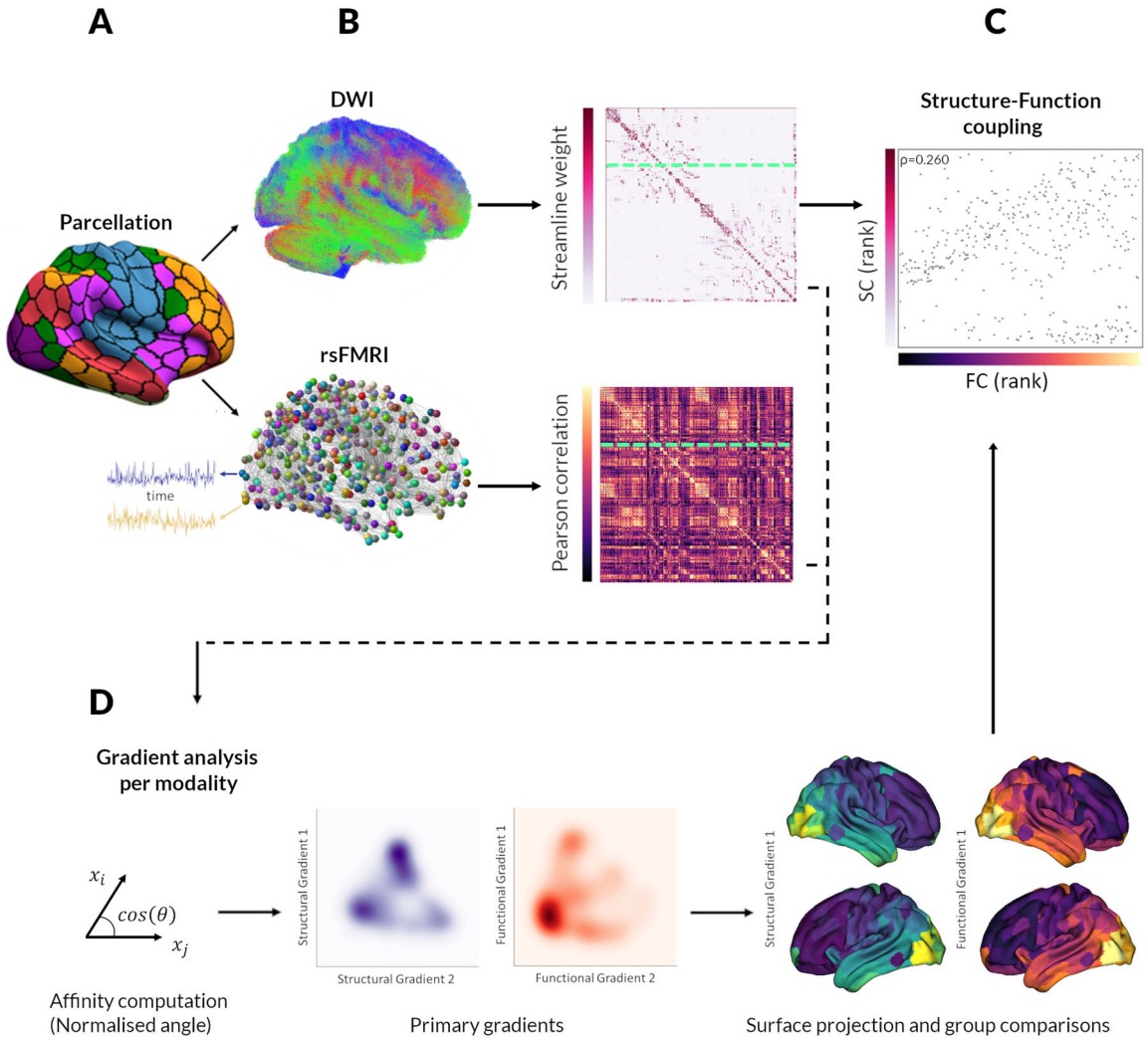

**Fig. 1 Overview of the study methodology. A** Analyses were conducted using a whole-brain parcellation including 400 cortical regions[42]. **B** Structural connectivity (SC) and functional connectivity (FC) matrices were derived for each participant from diffusion-weighted imaging (DWI) and resting-state functional MRI (rsfMRI) data, respectively. SC: Darker colours indicate higher normalised streamline counts; FC: lighter colours indicate higher Fisher-z normalised Spearman correlation values between every possible pair of brain regions. **C** For each participant, regional connectivity profiles were extracted from each row of the structural or functional connectivity matrix (example here shown by green dashed line) and represented as vectors of connectivity strength from a single network node to all other nodes in the network. Structural–functional connectivity coupling (SC–FC coupling) was then measured as the Spearman rank correlation between non-zero elements of regional structural and functional connectivity profiles. SC–FC coupling was then compared between groups. **D** Gradients of connectivity covariance were constructed for each individual's structural and functional connectivity matrices using diffusion map embedding, a non-linear compression algorithm that sorts nodes based on affinity (normalised angle was used as a measure of affinity). We focused our analyses on the first 2 principal structural and functional gradients; the scores of each node for the first 2 gradients are shown in the kernel density plot (blue: structural, red: functional gradients). Gradient scores and SC–FC coupling may be projected back to the cortical surface. We then correlated functional and structural gradient scores with SC–FC coupling for each region.

and middle occipital gyri and right cuneus, precuneus and calcarine gyrus (Fig. 2D and Table 2).

When we compared overall coupling, averaged across the whole of the brain network, PD low visual performers did not show significant decoupling compared to PD high visual performers (mean 0.469 in PD low visual performers vs 0.492 in PD high visual performers, $p = 0.415$) (Fig. 2C). In contrast, changes in PD low visual performers were more focal (Fig. 2B) with bilateral insula and the right calcarine gyrus showing significant decoupling compared to high visual performers (Fig. 2D and Table 2). Higher SC–FC coupling within the right calcarine gyrus was related to higher MOCA scores in PD participants ($r = 0.307$, $q = 0.011$) (Supplementary Fig. 3). There was no significant correlation between MOCA scores and SC–FC coupling in the left or right insula (left: $r = 0.099$, $q = 0.361$; right: $r = 0.062$, $q = 0.567$).

To ensure that results were not influenced by parcellation choice, we replicated our SC–FC analysis in another parcellation with similar results (Supplementary Figs. 4 and 5). Group differences in PD vs controls and PD low vs PD high visual performers in our cohort for structural and functional connectivity separately are found in Supplementary Fig. 6.

**Defining structural and functional gradients of macroscale cortical organisation in health.** Next, we assessed whether the spatial variability in structure–function decoupling aligns with fundamental properties of cortical organisation. Using diffusion map embedding for non-linear dimensionality reduction[46], we derived structural and functional gradients of cortical organisation for each control participant's structural and functional

**Table 1 Demographics and clinical assessments.**

| Characteristic | Controls n = 30 | PD high visual performers n = 58 | PD low visual performers n = 30 | Statistic |
|---|---|---|---|---|
| Age (years) | 66.8 (9.3) | **66.8 (9.3)** | **64.4 (8.0)** | **0.001**[b] |
| Male (%) | 13 (43.3) | 13 (43.3) | 48 (54.5) | 0.334 |
| Years of education | 17.7 (2.5) | 17.7 (2.5) | 17.1 (2.9) | 0.098 |
| Vision | | | | |
| Contrast sensitivity (Pelli-Robson)[a] | 1.8 (0.2) | **1.8 (0.2)** | **1.7 (0.1)** | **<0.001**[b] |
| Acuity (LogMar)[a] | −0.08 (0.2) | −0.08 (0.2) | −0.08 (0.1) | 0.441 |
| Colour vision (D15) | 1.3 (1.3) | 1.1 (0.8) | 1.6 (1.8) | 0.334 |
| General cognition | | | | |
| MOCA | 28.8 (1.3) | **28.8 (1.3)** | **27.9 (2.4)** | **0.003**[b] |
| MMSE | 29 (1.1) | 29 (1.1) | 28.9 (1.3) | 0.573 |
| Mood | | | | |
| HADS anxiety | **3.7 (3.4)** | **3.7 (3.4)** | **5.9 (3.1)** | **0.023**[b,c] |
| HADS depression | **1.9 (1.4)** | **1.9 (1.4)** | **4.1 (3.1)** | **<0.001**[b,c] |
| Disease-specific measures | | | | |
| Years from diagnosis | – | 3.7 (2.6) | 4.5 (2.6) | 0.061 |
| UPDRS total score | – | 44.2 (25.5) | 49.9 (26.8) | 0.315 |
| UPDRS motor score | – | 23. 2 (14.1) | 24.3 (14.8) | 0.785 |
| LEDD | – | 411.1 (273.5) | 479.4 (201.2) | 0.099 |
| RBDSQ | – | 4.4 (2.7) | 4.0 (2.1) | 0.832 |
| Image quality metrics | | | | |
| Coefficient of joint variation[d] | 0.69 (0.3) | 0.69 (0.3) | 0.67 (0.2) | 0.925 |
| Entropy focus criterion[d] | 0.59 (0.1) | 0.59 (0.1) | 0.59 (0.1) | 0.167 |
| Total Signal to noise ratio[e] | 1.89 (0.2) | 1.89 (0.2) | 1.85 (0.2) | 0.314 |
| Mean framewise displacement[e] | 0.17 (0.1) | 0.17 (0.1) | 0.19 (0.1) | 0.056 |

All data shown are mean (SD) except gender.
In bold characteristics that significantly differed between groups.
*HADS* Hospital anxiety and depression scale, *MMSE* Mini-mental state examination, *MOCA* Montreal cognitive assessment, *UPDRS* Unified Parkinson's Disease Rating Scale, *LEDD* Levodopa Equivalent Dose, *RBDSQ* REM sleep behaviour disorder scale.
[a]Best binocular score used; LogMAR: lower score implies better performance, Pelli–Robson: higher score implies better performance.
[b]Statistically significant difference between PD and controls.
[c]Statistically significant difference between PD high visual performers and PD low visual performers.
[d]Higher values imply worse image quality.
[e]Higher values imply better image quality.

connectivity matrix respectively. Similar to the previous studies[33,45,47,48], we focused our analyses on the first two principal gradients. The first principal gradient explained 14.3% of the variance for structural and 27.5% for functional gradients and the second principal gradient 11.9% for structural and 17.5% for functional gradients.

We assessed the dimension of variance in connectivity that the first two gradients represented in healthy controls. The first principal gradients (structural and functional) were anchored on one end in frontal and the other end on occipital regions (Fig. 3A: structural and Fig. 3B: functional gradients). To confirm this A–P alignment, we performed correlations (df = 400) between the weighting of each brain region in the first gradient (using the mean value across the control group only) and the corresponding A–P axis coordinate for that region. This showed a significant negative correlation for the first structural [$\rho = -0.626$ (interindividual range: $-0.651$, $-0.572$), $p_{spin} < 0.001$] and functional gradient [$\rho = -0.592$ (interindividual range: $-0.684$, $-0.267$), $p_{spin} < 0.001$] (Fig. 3A, B).

In contrast, the second principal gradients in control participants were anchored in unimodal regions (primary sensory cortex) on one end and transmodal regions on the other end (Fig. 3C structural and 3D functional gradients). To confirm this, we assigned each brain region to a level of hierarchy according to its corresponding functional network, moving from unimodal (level 1) to transmodal areas (level 4)[49]. We then performed correlations (df = 400) between the weighting of each brain region in the second principle gradient and its hierarchy level. Both the structural [$\rho = 0.478$ (interindividual range: 0.372, 0.518), $p_{spin} = 0.003$] and functional second principal gradients

[$\rho = 0.663$ (interindividual range: 0.239, 0.749), $p_{spin} = 0.001$] significant correlated with this unimodal–transmodal axis (Fig. 3C, D).

**Structure–function decoupling occurs across gradients of macroscale organisation in health and is accelerated in PD.** Next, we examined the relationship between macroscale gradients and SC–FC coupling using a spatial permutation test. This generates a null distribution of randomly rotated brain maps that preserve the spatial covariance structure of the original data (the resulting p-values are denoted $p_{spin}$)[50].

In controls, variation in SC–FC coupling significantly correlated with the first principal gradients, with stronger coupling in posterior regions and weaker in anterior ones (structural: $\rho = -0.169$, $p_{spin} = 0.011$, functional: $\rho = -0.2$, $p_{spin} = 0.042$; Fig. 4A, B). Coupling also significantly correlated with the second principal gradients: unimodal sensory regions exhibited relatively strong SC–FC coupling but transmodal regions exhibited weaker coupling (structural: $\rho = -0.144$, $p_{spin} = 0.007$, functional: $\rho = -0.203$, $p_{spin} = 0.009$; Fig. 4C, D).

This gradual decoupling in SC–FC across the A–P and unimodal–transmodal axes seen in controls, was amplified in PD and even more so in low visual performers. Greater SC–FC decoupling was seen along the A–P axis for both structural (PD high visual performers $\rho = -0.276$, $p_{spin} < 0.001$; PD low visual performers $\rho = -0.307$, $p_{spin} < 0.001$; Fig. 4A) and functional gradients (PD high visual performers $\rho = -0.271$, $p_{spin} < 0.001$; PD low visual performers $\rho = -0.349$, $p_{spin} < 0.001$; Fig. 4B). Similarly, greater decoupling was seen along the

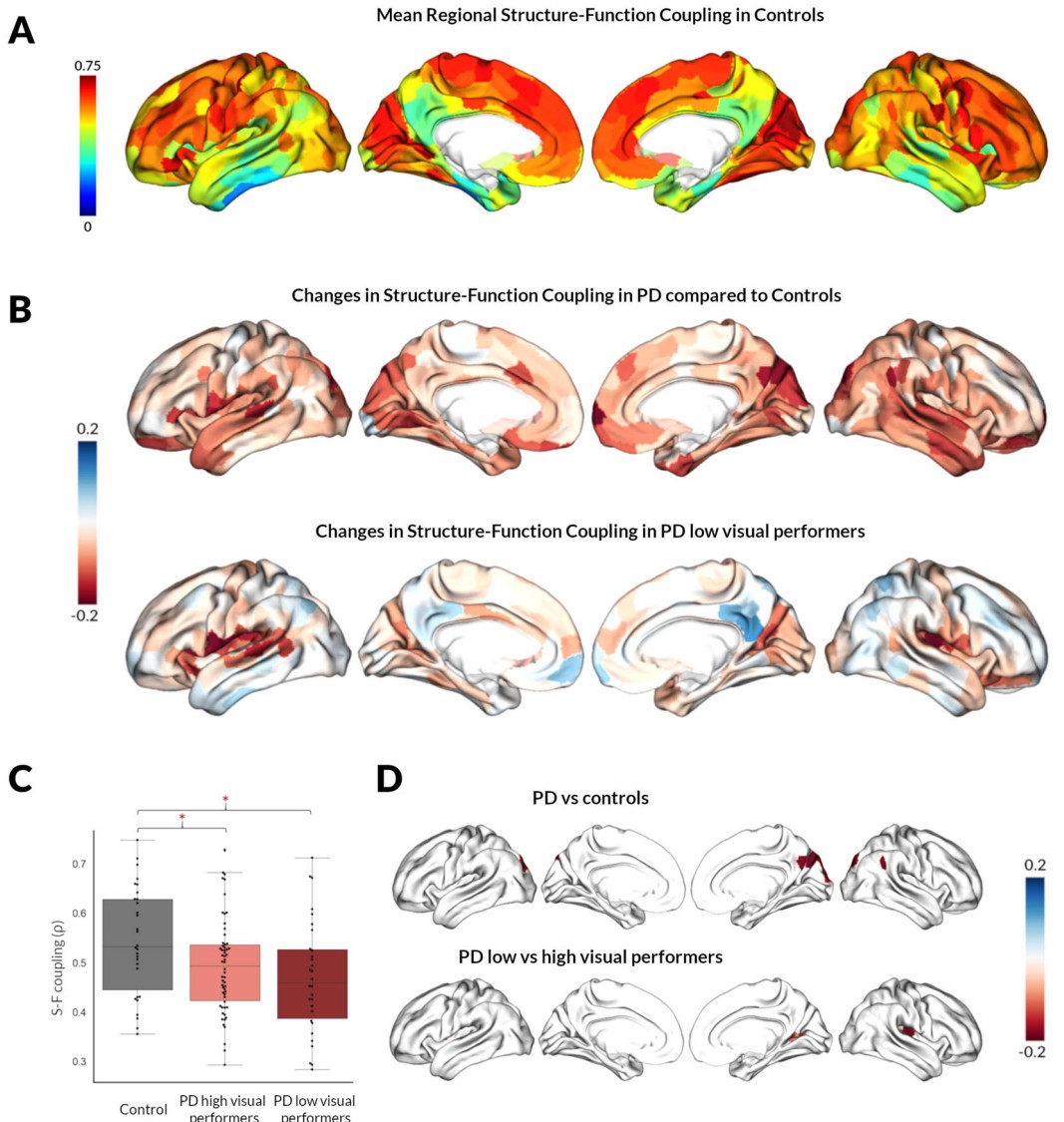

**Fig. 2 Structural–functional connectivity coupling in controls and changes in patients with Parkinson's disease (PD). A** Spatial pattern of structural–functional connectivity (SC–FC) coupling in controls. The coupling between regional structural and functional connectivity profiles varied widely across the cortex. Primary sensory and medial prefrontal cortex exhibited relatively high structure–function coupling, while lateral temporal and parietal regions showed relatively low coupling. **B** Spatial pattern of SC–FC decoupling in PD. Regional changes in SC–FC coupling (correlation coefficient plotted, with age and gender correction) are presented in PD vs controls (top) and PD low visual performers vs PD high visual performers (bottom). **C** SC–FC coupling changes averaged across all nodes of the network. Average SC–FC coupling (Spearman's rank correlation) across the whole-brain network (400 nodes) is compared between controls, PD high and PD low visual performers. S–F: structural connectivity–functional connectivity. * denotes statistically significant results ($p$-spin < 0.05). Both PD low and PD high visual performers showed significantly reduced global coupling than controls (PD low visual performers mean 0.469 vs 0.544 in controls, $p = 0.002$; PD high visual performers mean 0.492 vs 0.544 in controls, $p = 0.005$). There was no significant difference between PD low and PD high visual performers ($p = 0.415$). **D** SC–FC coupling changes for each node across the brain. Whole-brain comparisons of SC–FC coupling were performed for every node across the whole brain between PD vs controls (top) and PD low visual performers vs PD high visual performers (bottom), age and gender included as covariates. Only nodes surviving FDR correction ($q < 0.05$) are presented.

unimodal–transmodal axis (structural: PD high visual performers $\rho = -0.161$, $p_{\text{spin}} = 0.040$; PD low visual performers $\rho = -0.207$, $p_{\text{spin}} = 0.001$; Fig. 4C and functional gradients: PD high visual performers $\rho = -0.241$, $p_{\text{spin}} = 0.005$; PD low visual performers $\rho = -0.268$, $p_{\text{spin}} = 0.001$; Fig. 4D).

**Relationship between structural–functional connectivity decoupling in PD and neurotransmitter receptor gene expression.** Finally, to assess the role that neuromodulatory systems may have in SC–FC decoupling in PD, we investigated the relationship between maps of gene expression for neurotransmitter receptor

genes (derived from post-mortem human brains) and SC–FC coupling changes in: (1) PD vs controls and (2) PD low vs high visual performers. We found that decoupling in PD showed a statistically significant moderate correlation with regional differences in gene expression of dopaminergic, serotoninergic and cholinergic receptors (Fig. 5A and Table 3). Specifically, decoupling in PD was associated with reduced expression of *DRD2* and three serotonin receptors (*HTR2A, HTR2C, HTR4*) and increased expression of a cholinergic (*CHRNA4*) and serotoninergic receptor (*HTR1E*) (Table 3).

In contrast, changes in SC–FC coupling in PD low visual performers (compared to high visual performers) were not

**Table 2 Nodes showing structural-functional connectivity decoupling in PD compared with controls and in PD low vs high visual performers.**

**Reduced SC-FC coupling in PD vs controls**

| Region | Coordinates in MNI space | | | SC–FC coupling in controls | SC–FC coupling in PD | Network | q value |
|---|---|---|---|---|---|---|---|
| | x | y | z | | | | |
| Superior occipital gyrus L | −16 | −89 | 33 | 0.601 (0.169) | 0.458 (0.177) | Visual | 0.026 |
| Middle occipital gyrus L | −26 | −70 | 31 | 0.659 (0.106) | 0.564 (0.144) | DAN | 0.050 |
| Middle occipital gyrus R | 43 | −79 | 10 | 0.647 (0.147) | 0.509 (0.181) | Visual | 0.034 |
| Calcarine R | 16 | −66 | 19 | 0.738 (0.053) | 0.615 (0.178) | Visual | 0.050 |
| Cuneus R | 14 | −78 | 34 | 0.706 (0.043) | 0.541 (0.175) | Visual | 0.005 |
| Superior occipital gyrus R | 16 | −87 | 36 | 0.659 (0.148) | 0.506 (0.187) | Visual | 0.014 |
| Angular gyrus R | 53 | −53 | 26 | 0.630 (0.145) | 0.462 (0.222) | DMN | 0.040 |
| Precuneus R | 6 | −52 | 23 | 0.418 (0.239) | 0.362 (0.203) | DMN | 0.040 |

**Reduced SC-FC coupling in PD low vs high visual performers**

| Region | Coordinates in MNI space | | | SC–FC coupling in PD high visual performers | SC–FC coupling in PD low visual performers | Network | q value |
|---|---|---|---|---|---|---|---|
| | x | y | z | | | | |
| Insula L | −36 | 4 | 11 | 0.355 (0.237) | 0.163 (0.288) | VAN | 0.012 |
| Calcarine R | 22 | −59 | 6 | 0.586 (0.125) | 0.480 (0.164) | Visual | 0.012 |
| Insula R | 35 | −21 | 14 | 0.588 (0.198) | 0.392 (0.114) | Sensorimotor | 0.012 |

*SC–FC coupling* structural connectivity-functional connectivity coupling. Shown as mean (std).
*DAN* dorsal attention network, *DMN* default mode network, *VAN* ventral attention network, *L* left, *R* right.

significantly correlated with changes in dopaminergic but rather to cholinergic (*CHRNA2, CHRNA3, CHRNA4*), serotoninergic (*HTR1A, HTR5A*) and noradrenergic (*ADRA2A*) receptors (Fig. 5B, Table 3 and Supplementary Table 3 for the full neurotransmitter gene expression results.).

## Discussion

We provide evidence of significant differences in SC–FC coupling in patients with PD and shed light onto the organisational and neuromodulatory principles that drive this decoupling.

In patients with PD, we found a spatially widespread decoupling of SC–FC correlations. In contrast, PD low visual performers, who are at higher risk of dementia, exhibited more focal decoupling compared to PD high visual performers, with the insula preferentially affected. SC–FC decoupling in PD follows specific gradients of hierarchical organisation: anterior–posterior and unimodal–transmodal. These same gradients governed spatial variation in SC–FC coupling in healthy controls but became more pronounced in PD and even more so in PD low visual performers.

We found that structural–functional connectivity decoupling in PD follows a unimodal-to-transmodal gradient. Several studies in health have shown stronger SC–FC coupling in unimodal sensory cortex and relative decoupling in transmodal association cortex coinciding with improvements in executive ability and abstract reasoning[10,11,33]. Our second principal gradients similarly reflected a unimodal-to-transmodal hierarchy and were correlated with SC–FC in controls. This provides further support to the tethering hypothesis that association cortex is untethered from molecular gradients of early sensory cortex[51], now using for the first time, gradients derived from diffusion-weighted imaging.

We show that in PD, structural and functional connectivity became more decoupled in regions higher along the unimodal–transmodal hierarchy. This supports the central role of the DMN in PD-associated cognitive impairment which has been highlighted by rsfMRI studies[52–54], pathological evidence[55] and, more recently, network lesion mapping[56]. Transmodal regions, such as the DMN, where SC–FC are, normally, less closely aligned may be more

vulnerable to the presence of neurodegeneration. Decoupling in these higher-order regions could explain the higher prevalence of neurocognitive deficits seen in PD, such as hallucinations and delusions, with a release of these regions from the normal constraints of sensory processing. Although in health a weaker SC–FC coupling may be beneficial allowing for more adaptive and flexible cognition, in the presence of neurodegeneration it may make transmodal regions more vulnerable. The numbers of patients with hallucinations and delusions in our cohort were too low to formally test whether these symptoms correlate with greater decoupling, but this would be an avenue of interest for future work.

In addition, we saw a striking increase of SC–FC decoupling along the A–P axis (first principle gradients) in PD. This correlation was observed in controls but became more pronounced in PD and even more so in low visual performers. An anterior–posterior spatial gradient has been observed at the gene expression level in the adult human brain[57–59] and prenatally[60,61]. Specific gene expression patterns across this gradient could confer vulnerability in the presence of degeneration. The A–P gradient however does not only reflect transcriptional differences but also changes in cortical microstructure with increase in neuronal number and density and decrease in neuron and arbour size across the A–P axis[58]. The increased neuronal population in more posterior regions may make them more vulnerable to transneuronal alpha-synuclein spread.

Finally, we shed light onto the neuromodulatory systems associated with SC–FC coupling in PD overall and in those individuals at higher risk of cognitive decline. Unsurprisingly, the reduced dopaminergic transmission was associated with SC–FC decoupling observed in PD compared to controls. In contrast, we found no correlation of dopaminergic receptor expression and decoupling in PD low visual performers, suggesting that neuromodulators other than dopamine may have a more important role in the development of cognitive impairment.

Altered serotoninergic transmission was also associated with SC–FC decoupling in PD participants, in keeping with evidence from positron emission tomography[62], biochemical[63] and

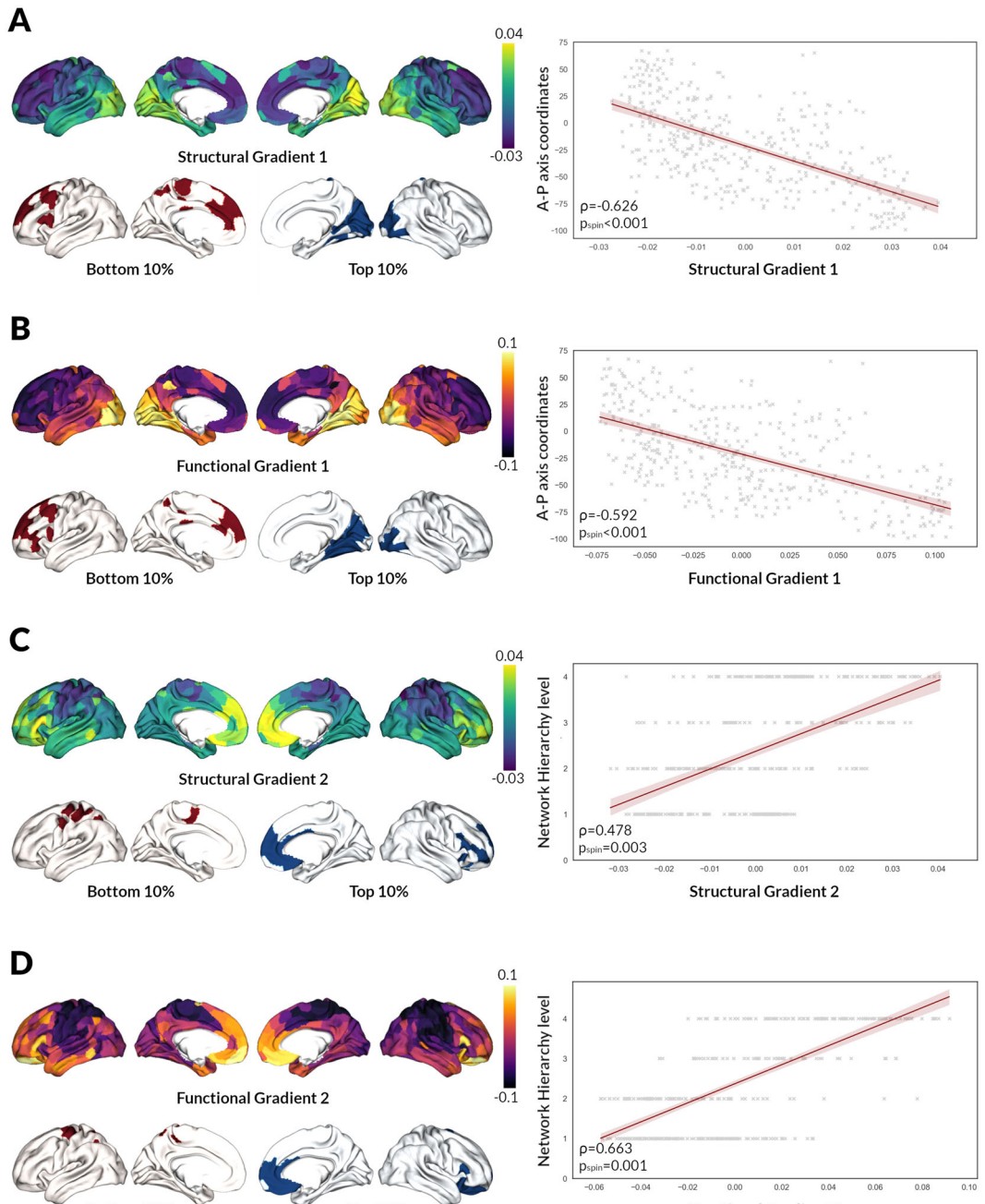

**Fig. 3 Structural and functional gradients of cortical organisation in controls.** The first two principal gradients derived from the averaged control structural and functional connectivity matrices are presented. Gradient scores are projected back onto the cortical surface. The first principal structural (**A**) and functional (**B**) gradients showed a dissociation between the posterior and anterior regions. The second principal structural (**C**) and functional (**D**) gradients showed a dissociation between unimodal and transmodal regions. Top and bottom 10% of the average control gradients highlight regions with similar (same colour) and distinct (red vs blue) connectivity profiles. For the first structural and functional gradients, top 10% of regions are more posterior and bottom 10% more anterior. For the second structural and functional gradients, top 10% of regions are more transmodal and bottom 10% more unimodal. On the right, we plot the correlation between the gradient score (control-averaged) and the A–P axis coordinate for the first principal gradients, and the Network Hierarchy level for the second principal gradients (each dot represents a single region of the average control connectome). A–P: Anterior–Posterior (lower values representing more posterior regions, higher values more anterior regions), Network Hierarchy level: level 1 sensory and sensorimotor networks, level 2 dorsal attention and salience networks, level 3 frontoparietal and limbic networks, level 4 default mode network (DMN).

post-mortem studies[64] showing serotoninergic degeneration in PD. In contrast, in PD low visual performers SC–FC decoupling was more prominent in regions with increased serotoninergic receptor expression, specifically *HTR1E* and *HTR5A*. Although the function of these receptors is not yet fully described, HTR5A is thought to have a role in cognition[65], with 5HT-5a antagonists improving cognition in animal models[66].

In addition, regional differences in nicotinic cholinergic receptors were associated with SC–FC decoupling with changes in both PD overall and PD low visual performers. Cholinergic cell involvement is well recognised in PD and linked to the development of dementia, with a progressive reduction in nicotinic receptors in parallel to dementia severity[67]. This reduction in PD could be more prominent in regions typically rich in nicotinic receptors in health.

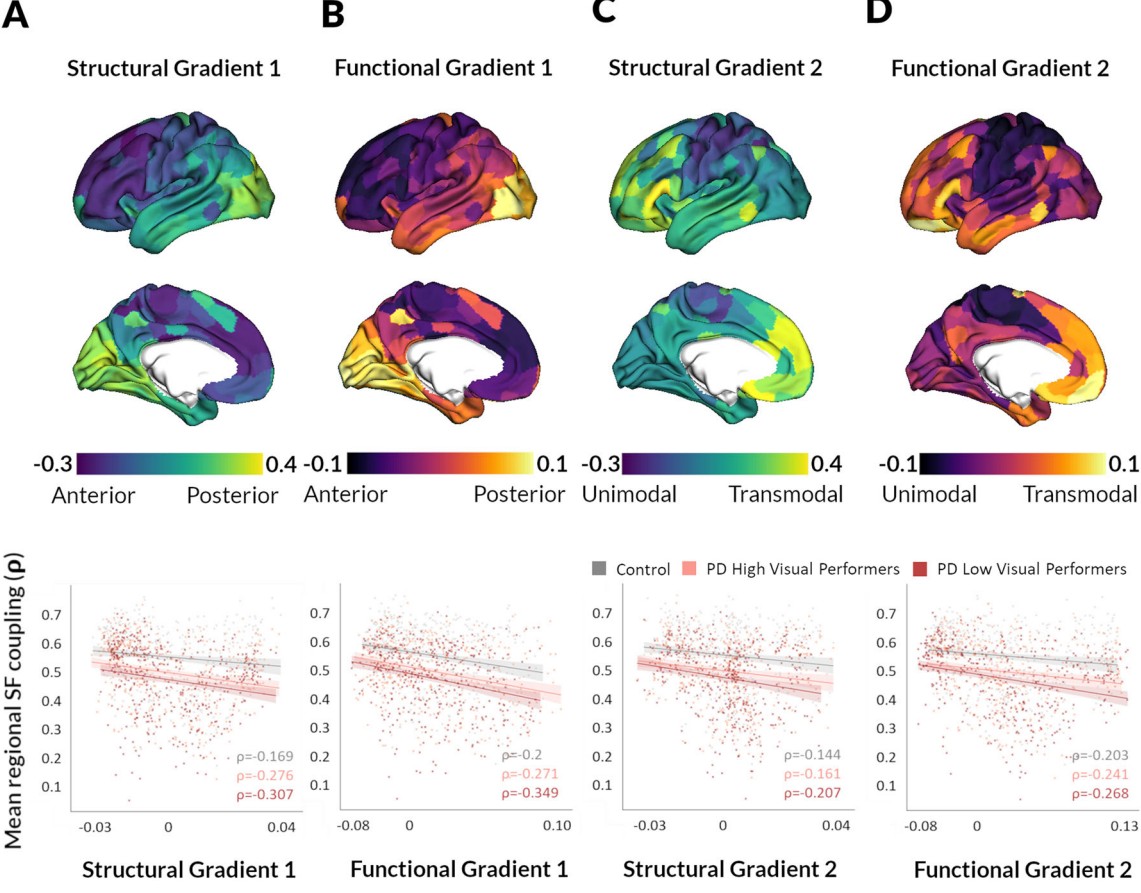

**Fig. 4 Structural–functional connectivity decoupling in PD follows macroscale cortical gradients.** Structural–functional connectivity (SC–FC) coupling is significantly associated with the first principal structural (**A**) and functional gradients (**B**), which align with the anterior–posterior axis (visualised on the top: lower gradient values represent more anterior regions, higher gradient values more posterior regions). The correlation between mean SC–FC coupling and gradient value is plotted for each brain region in controls (grey), PD high visual performer (pink) and PD low visual performers (red) with $\rho$ denoting the Spearman correlation coefficient. This correlation was seen in all groups but was more pronounced in PD than control participants and even more so in PD low visual performers (who are at higher risk of dementia). SC–FC coupling also reflected a brain region's position along the second principal structural (**C**) and functional gradients (**D**), which reflect a unimodal-to-transmodal axis (visualised on the top: lower gradient values represent more unimodal regions, higher gradient values more transmodal regions). The correlation between mean SC–FC coupling and gradient value is plotted for each brain region in controls (grey), PD high visual performer (pink) and PD low visual performers (red) with $\rho$ denoting the Spearman correlation coefficient. Again this relationship was more pronounced in PD low visual performers then PD high visual performers followed by control participants. The significance of regional correlations was evaluated using nonparametric spatial permutation testing.

Finally, we found that SC–FC decoupling in PD low visual performers was more pronounced in regions with reduced expression of the noradrenergic receptor *ADRA2* in health ($q = 0.041$). Interestingly, ADRA2 gene polymorphisms were recently identified in a genome-wide association study of PD patients (associated with increased insomnia at baseline)[68]. Norepinephrine and its receptors have also been linked to PD[69–71], although not previously in relation to cognitive impairment.

Several methodological considerations need to be taken into account when interpreting the results of our study. Structural connectivity was estimated using streamlines from diffusion tractography which is susceptible to false positives and false negatives[72]. To provide the best possible estimate of structural connectivity, we used multi-shell data and improved post-processing, including constrained spherical deconvolution[73] and SIFT2[74]. Functional connectivity estimates were derived from rsfMRI data which are also susceptible to the artefact, particularly motion. To mitigate this, we adopted rigorous quality assurance and strict exclusion criteria[75]. Time of day and medication usage influence rsfMRI[76]; all participants were scanned in the ON state, receiving their usual dopaminergic medications and at the same

time. Although we optimised both our structural and functional connectivity estimates, these remain indirect measures of brain structure and function which needs to be taken into account when interpreting the results of our study. We used parcellated data to allow for group comparisons, however functional boundaries vary across individuals[77] which could lead to misalignments when comparing structural–functional connectivity relationships. We used gene expression data from healthy human brains, therefore results relating to neurotransmitter receptor gene expression should be interpreted with caution. In addition, although significantly correlated, regional variation in gene expression explained only a moderate fraction of the variance in SC–FC coupling (absolute value of correlation coefficients between 0.133 and 0.308), suggesting that additional factors other than neurotransmitter receptor gene expression have a role in the changes in SC–FC coupling in PD. However, our study could provide insights informing subsequent validation studies in PD brains or animal models. Finally, our study examines cross-sectional data, using visual dysfunction as a surrogate marker for dementia risk. Although this provides useful insights, longitudinal studies in PD patients who progress to dementia are likely

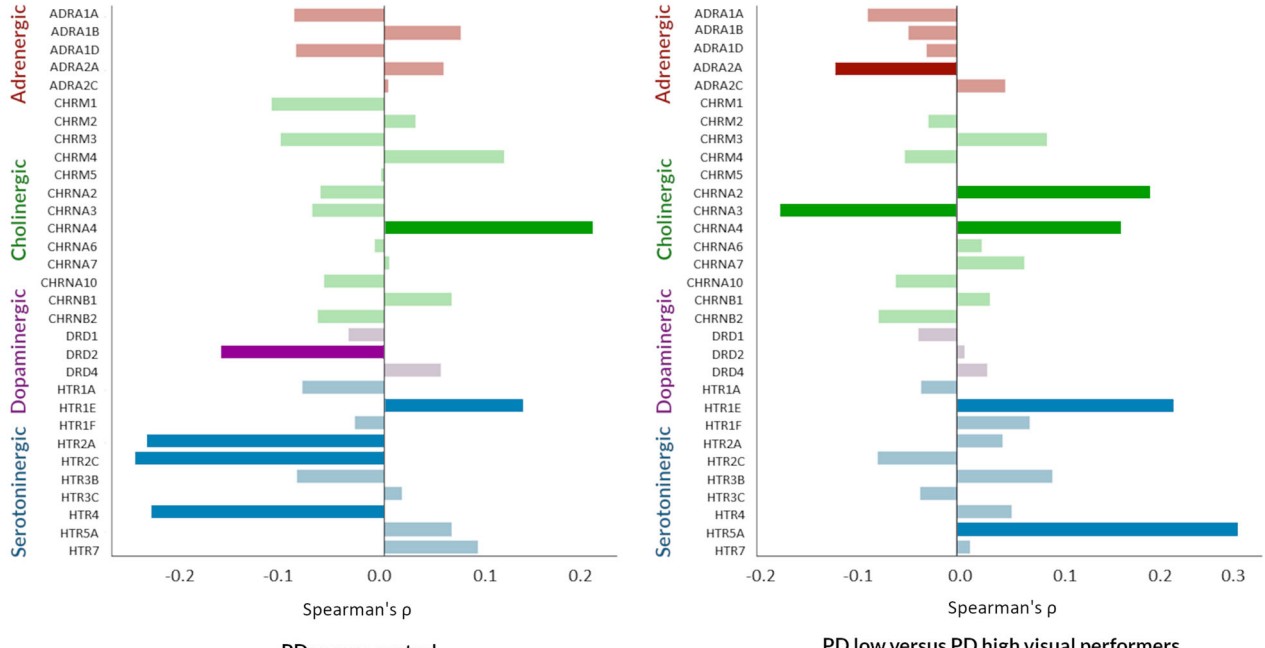

**Fig. 5 Correlation between regional cortical expression of neurotransmitter receptor genes and structural–functional connectivity decoupling in PD.** Spearman correlations between regional cortical expression of adrenergic, cholinergic (muscarinic and nicotinic), and dopaminergic receptors and difference in structural–functional connectivity coupling seen between PD and controls (left) and PD low visual performers vs PD high visual performers (right). Full gene names in Supplementary Table 2. Results colour coded according to receptors: red: adrenergic, green: cholinergic, purple: dopaminergic, blue: serotoninergic receptors. Bars with stronger (rather than fainter) colours indicate statistically significant relationships (FDR-corrected $p > 0.05$).

**Table 3 Neurotransmitter receptor genes correlating with the change in structural–functional connectivity coupling in PD.**

**Genes correlated with structural-functional connectivity decoupling in PD compared to controls**

| Gene symbol | Ligand | Correlation coefficient | $q$ value |
|---|---|---|---|
| CHRNA4 | Acetylcholine | 0.210 | <0.001 |
| DRD2 | Dopamine | −0.165 | 0.006 |
| HTR1E | Serotonin | 0.140 | 0.027 |
| HTR2A | Serotonin | −0.239 | <0.001 |
| HTR2C | Serotonin | −0.251 | <0.001 |
| HTR4 | Serotonin | −0.234 | <0.001 |

**Genes correlated with structural-functional connectivity decoupling in PD low visual performers compared to PD high visual performers**

| Gene symbol | Ligand | Correlation coefficient | $q$ value |
|---|---|---|---|
| ADRA2A | Norepinephrine | −0.133 | 0.041 |
| CHRNA2 | Acetylcholine | 0.211 | <0.001 |
| CHRNA3 | Acetylcholine | −0.194 | 0.001 |
| CHRNA4 | Acetylcholine | 0.180 | 0.002 |
| HTR1E | Serotonin | 0.237 | <0.001 |
| HTR5A | Serotonin | 0.308 | <0.001 |

Note that correlation coefficients of absolute values between 0.1 and 0.4 represent moderate correlation in our dataset. In all cases, gene expression levels were significant in spatial permutation testing ($p$-spin < 0.001).
*CHRNA* Nicotinic Cholinergic Receptor (Alpha), *DRD* Dopamine Receptor D, *HTR* 5-Hydroxytryptamine Receptor, *ADRA* Alpha-1A adrenergic receptor.

to provide further insights into the temporal order of structural–functional connectivity decoupling in PD.

Our findings show that structural–functional connectivity coupling is severely disrupted in PD across the cortex, with even more pronounced decoupling in temporal lobe structures in low visual performers (who are at higher risk of dementia). We show that structural–functional connectivity decoupling in PD follows the same macroscopic organisational principles that guide SC–FC coupling in healthy individuals but with accelerated decoupling. Finally, we clarify the neuromodulatory correlates of SC–FC

decoupling in PD. Altogether, our findings propose a framework to explain SC–FC decoupling in PD and offer insights to possible therapeutic targets.

## Methods
**Participants**. We included 88 patients with PD and 30 unaffected controls, recruited to our London centre. All patients with PD fulfilled the Queen Square Brain Bank Criteria[78]. All participants with diffusion-weighted imaging and rsfMRI scans passing predefined quality control criteria (see "Methods: Data acquisition & Quality assurance" section) were included. The study was approved by the local ethics committee and participants provided written informed consent.

 9

Participants with PD were classified according to their performance in two computer-based higher-order visual tasks. The Cats and Dogs task measures tolerance to visual skew, with images of cats and dogs distorted by varying skew along the X axis and threshold of visual skew determined using psychophysical testing (two-alternative forced-choice, 90 repetitions) (as described previously[4,7,43] and see example stimulus in Supplemental Fig. 1). The biological motion task measures sensitivity to the perception of a moving person from moving dots at the position of the major joints. Increasing the number of moving dots makes the task more difficult, and the number of additional dots tolerated is determined psychophysically, as previously described[44] and see Supplemental Fig. 1 for example stimulus. These visual tasks have been chosen as they provide robust measures of higher-order visual function and have been shown by our group to be associated with a higher risk of PD dementia and worsening cognition over time[4,7,44].

To capture patients with consistently poor performance in these high-level visual tasks, we classified patients as poor visual performers if they performed worse than the group median in both tasks ($n = 33$ low visual performers). All other patients with PD were classified as high visual performers ($n = 58$) as in the previous work[4,79,80]. Details on task performance in the two experimental tasks are seen in Supplementary Fig. 2. Thirty unaffected age-matched controls were recruited from spouses and a volunteer database; controls were matched to the PD group as a whole.

The Mini-Mental State Examination (MMSE)[81] and Montreal Cognitive Assessment (MoCA)[82] were used as measures of general cognition. Additionally, two tests per cognitive domain were performed[83]: Digit span[84] and Stroop colour[85] for attention, Stroop interference[85] and Category fluency[86] for executive function, Word recognition task[87] and Logical memory[84] for memory, Graded naming task[88] and Letter fluency[86] for language, and Visual object and space perception battery[89], and Hooper visual organisation test[90] for visuospatial function. Visual acuity was assessed using LogMAR[91], colour vision using Farnsworth D15[92], and contrast sensitivity using Pelli–Robson[93]. The Hospital Anxiety and Depression Scale (HADS) was used to assess mood[94]. PD participants underwent assessments of motor function using MDS-UPDRS[95], sleep using the REM Sleep Behaviour Disorder Questionnaire[96] and smell using Sniffin' Sticks[97]. Levodopa dose equivalence scores (LEDD) were calculated for PD participants[98].

**Data acquisition and quality assurance**. All MRI data were acquired on a 3T Siemens Magnetom Prisma scanner (Siemens) with a 64-channel head coil. Diffusion-weighted imaging (DWI) was acquired with the following parameters: b0 in both AP and PA directions, $b = 50$ s/mm$^2$/17 directions, $b = 300$ s/mm$^2$/8 directions, $b = 1000$ s/mm$^2$/64 directions, $b = 2000$ s/mm$^2$/64 directions, $2 \times 2 \times 2$ mm isotropic voxels, TE = 3260 ms, TR = 58 ms, 72 slices, 2 mm thickness, acceleration factor = 2. DWI acquisition time was ~10 min. Resting-state functional MRI (rsfMRI) was acquired with the following parameters: gradient-echo EPI, TR = 70 ms, TE = 30 ms, flip angle = 90°, FOV = 192 × 192, voxel size = 3 × 3 × 2.5 mm, 105 volumes, 7-min session. During rsfMRI, participants were instructed to lie quietly with their eyes closed and avoid falling asleep; this was confirmed by monitoring and post-scan debriefing. A 3D MPRAGE (magnetisation prepared rapid acquisition gradient-echo) image (voxel size = 1 × 1 × 1 mm, TE = 3.34 ms, TR = 2530 ms, flip angle = 7°) was also obtained. Imaging for all participants was performed at the same time of day, with PD participants receiving their normal medications.

Both modalities underwent rigorous quality assurance. Prior to diffusion processing, all volumes of raw datasets were visually inspected and each volume evaluated for the presence of artefact; only scans with <15 volumes containing artefacts[99] were included. As a result, 3 PD and 1 control participants were excluded from the original patient cohort.

Quality of rsfMRI data was assessed using the MRI Quality Control tool[100]. As rsfMRI can be particularly susceptible to motion effects we adopted stringent exclusion criteria[75]. Specifically, participants were excluded if any of the following was met: (1) mean framewise displacement (FD) > 0.3 mm, (2) any FD > 5 mm, or (3) outliers >30% of the whole sample. This led to 12 participants being excluded (11 PD, of whom 5 low visual performers, and 1 control), resulting in 88 patients included in the dataset presented here.

**Parcellation**. An overview of the study methodology is seen in Fig. 1. 400 cortical regions of interest (ROIs) were generated by segmenting each participant's T1-weighted image using the Schaefer parcellation[42]. We replicated SC–FC coupling analyses using the Glasser parcellation[101]. Parcellations over 200 nodes increase reliability in gradient construction, particularly those derived from functional connectivity[102]. We used the same parcellation to construct functional and structural connectivity matrices for each participant (Fig. 1A).

**Structural connectome construction**. Pre-processing of DWI images was performed in MRtrix3.0[103]. Diffusion-weighted images underwent denoising[104], removal of Gibbs artefacts[105], eddy-current and motion correction[106], and bias field correction[107]. Diffusion tensor metrics were calculated and constrained spherical deconvolution performed[108]. The raw T1-weighted images were registered to the diffusion-weighted image using NiftyReg[109] and five-tissue anatomical segmentation performed using the 5ttgen script in MRtrix.

Subsequently, we performed anatomically constrained tractography with 10 million streamlines[110] using the iFOD2 tractography algorithm[111] and dynamic seeding with streamlines truncated at the grey-white matter interface. We applied the spherical deconvolution informed filtering of tractograms (SIFT2) algorithm[74] to reduce biases. The resulting set of streamlines was used to construct the structural brain network. Connections were weighted by streamline count and a cross-sectional area multiplier[74] and combined to a 400 × 400 undirected, weighted matrix (Fig. 1B). As recommended by the authors of SIFT2, we did not apply a threshold to structural connectivity matrices[74].

**Functional connectome construction**. rsfMRI data underwent standard pre-processing using fMRIPrep 1.5.0[112]. The first 4 volumes were discarded to allow for steady-state equilibrium. Functional data was slice-time corrected using 3dTshift from AFNI[113] and motion-corrected using mcflirt[114]. Distortion correction was performed using a TOPUP implementation[115]. This was followed by co-registration to the corresponding T1-weighted image using boundary-based registration with six degrees of freedom[116]. Motion correcting transformations, field distortion correcting warp, BOLD-to-T1w transformation and T1w-to-template (MNI) warp were concatenated and applied in a single step using antsApplyTransforms (ANTs v2.1.0) using Lanczos interpolation.

Physiological noise regressors were extracted applying CompCor[117]. Sources of spurious variance were removed through linear regression (six motion parameters, mean signal from white matter and cerebrospinal fluid), followed by calculation of bivariate correlations and application of Fisher transform. Given the contentiousness of global signal regression[118] and potential to distort group differences[119], we did not regress global signal.

Functional connectivity between ROIs was quantified as the Pearson correlation coefficient between mean regional BOLD time series. To minimise the effect of spurious connections whilst avoiding arbitrary thresholds, we used structural connectivity to inform functional connectome construction. Specifically, we discarded functional connections between ROIs that were solely based on time series correlation in the absence of anatomical connection. For each participant, a 400 × 400 weighted adjacency matrix was constructed representing the functional connectome (Fig. 1B).

**Structural–functional connectivity coupling analysis**. We extracted regional connectivity profiles for each participant's structural and functional connectivity matrix, as vectors of connectivity strength from a single node to all other nodes in the network. SC–FC coupling for each node was then measured as the Spearman rank correlation between the non-zero elements of the regional structural and functional connectivity profiles[10,120,121] (Fig. 1C).

**Gradient analysis**. We derived cortical gradients separately from structural and functional connectivity matrices, using diffusion map embedding. This identifies spatial axes of variation in connectivity across different areas, whereby cortical vertices that are strongly interconnected are closer together and vertices with little or no inter-connectivity are farther apart[45,46]. We used normalised angle as a metric of similarity (values between 0 and 1, with 1 denoting identical angles, and 0 opposing angles). The normalised angle between two nodes $i$ and $j$ ($A(i, j)$) is calculated as shown in equation 1 below:

$$A(i, j) = 1 - \frac{\cos^{-1}\left(\cos \text{sim}\left(x_i, x_j\right)\right)}{\pi} \qquad (1)$$

where cos sim is the cosine similarity function. First, we generated a group-level gradient component template from the average structural and functional connectivity matrices of all participants. We performed Procrustes alignment to align the gradient components of each individual to the group template[122]. Gradient components defined in connectivity space were mapped back onto the cortical surface (Fig. 1D). For each derived gradient, we calculated the variance explained by dividing the gradient's eigenvalue with the sum of the eigenvalues for all gradients[102]. Gradient analyses were performed using BrainSpace[102].

To assess the correspondence of the first structural and functional gradients with the A–P axis, we calculated the correlation between A–P axis coordinates for each brain region[42] and its corresponding gradient coefficient. To ensure that the second structural and functional gradients represented a unimodal–transmodal gradient we assigned functional communities to levels of hierarchy (level 1: sensory and sensorimotor networks, level 2: dorsal attention and salience networks, level 3: frontoparietal and limbic networks, level 4: default mode network (DMN)[45,47,49]. We then calculated the Spearman correlation coefficient between a node's level of hierarchy and gradient coefficient.

**Neurotransmitter receptor gene expression**. Expression profiles for genes of noradrenergic, cholinergic (nicotinic and muscarinic), dopaminergic and serotoninergic receptors were obtained using data from the Allen Human Brain Atlas (AHBA)[57]. We used the recently described rigorous method of pre-processing by Arnatkevičiūtė et al.[123] to extract gene expression data from AHBA and map them to the 400 cortical regions of our parcellation, using abagen[124]. Each tissue sample was assigned to an anatomical structure of the 400 cortical regions, using the

AHBA MRI data for each donor. Data were pooled between homologous cortical regions to ensure adequate coverage of both the left (data from six donors) and right hemisphere (data from two donors). Distances between samples were evaluated on the cortical surface with a 2 mm distance threshold. Probe to gene annotations were updated in Re-Annotator[125]. Only probes where expression measures were above a background threshold in more than 50% of samples were selected. A representative probe for a gene was selected based on highest intensity. Gene expression data were normalised across the cortex using scaled, outlier-robust sigmoid normalisation. 15,745 genes (of 20,737 initially included in the Allen atlas gene expression data) survived these pre-processing and quality assurance steps. Expression profiles for 31 pre-selected genes (Supplementary Table 2) encoding receptors for norepinephrine, acetylcholine, dopamine and serotonin were then extracted for each of the 400 cortical regions of our parcellation.

**Statistics and reproducibility**. Demographics, clinical and imaging characteristics were compared between PD high visual performers, low visual performers and controls using ANOVA for normally distributed and Kruskal–Wallis for non-normally distributed variables (Shapiro–Wilk test for normality), with post-hoc testing using $t$-tests and Mann–Whitney respectively. Statistical significance defined as $p < 0.05$.

For group comparisons between SC–FC coupling and gradient component scores we used general linear model, with age and gender as covariates and comparisons of interest: (1) PD vs controls and (2) PD low visual performers vs PD high visual performers. We controlled for multiple comparisons using the False Discovery Rate (Benjamini–Hochberg method, $q < 0.05$) across 400 nodes.

The significance of correspondence between SC–FC coupling and gradient coefficients was estimated using a spatial permutation test, which generates randomly rotated brain maps whilst preserving spatial covariance[50]. We performed 1000 random spatial permutations[126] and calculated Spearman correlation coefficient between extracted regional SC–FC values and gradient coefficient to build a null distribution. The permutation-based $p$-value ($p_{spin}$) was calculated as the proportion of times that the null correlation coefficients were greater than the empirical coefficients[50,126].

Spearman correlations were performed between regional differences in SC–FC coupling between (1) PD vs controls and (2) PD low vs high visual performers. This was expressed as the vector of the difference in SC–FC coupling between groups (PD vs controls and PD low visual performers vs PD high visual performers for each of the 400 cortical nodes), visualised in Fig. 2B, and the regional expression level of each of the chosen 31 neurotransmitter receptor genes at each of the 400 cortical nodes. Results were FDR-corrected for multiple comparisons, $q < 0.05$, across 31 genes. Spatial permutation testing, as described above (1000 spatial permutations of the SC–FC regional differences for both PD vs controls and PD low vs PD high visual performers) were performed to ensure that the correlation between gene expression levels and SC–FC coupling was higher than expected by chance and had not arisen spuriously due to spatial autocorrelation[127]. Analyses were performed in Python 3 (Jupyter Lab v1.2.6).

**Reporting summary**. Further information on research design is available in the Nature Research Reporting Summary linked to this article.

## Data availability
Imaging and clinical data used in this study will be shared upon reasonable request to the corresponding author. All data and statistics generated from this study are presented in the manuscript and Supplementary Data 1–5.

## Code availability
All methods used open source software, and all links to the relevant software are included in Supplementary Methods (URLs). Code used in the analyses described in this paper is available here: https://github.com/AngelikaZa/SCFC.

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

## Acknowledgements

A.Z. is supported by an Alzheimer's Research UK Clinical Research Fellowship (2018B-001). P.M.C. is supported by the National Institute for Health Research. G.R. is supported by the Wellcome Trust. R.S.W. is supported by a Wellcome Clinical Research Career Development Fellowship (201567/Z/16/Z). We gratefully acknowledge the support of NVIDIA Corporation with the donation of the Quadro P6000 GPU used for this research. We acknowledge the use of the UCL Myriad High Performance Computing Facility (Myriad@UCL), and associated support services, in the completion of this work. This research was also supported by the National Institute for Health Research University College London Hospitals Biomedical Research Centre.

## Author contributions

A.Z.: conceptualisation, data curation, methodology, software, investigation, formal analysis, writing—original draft, writing—review & editing, visualisation. P.M.C.: conceptualisation, methodology, resources, writing—review and editing. A.J.L.: project administration, data curation, writing—review and editing. A.J.L.: conceptualisation, methodology, writing—review & editing. G.R.: conceptualisation, methodology, writing—review and editing. R.S.W.: conceptualisation, methodology, funding acquisition, project administration, supervision, writing—review and editing.

## Competing interests

R.S.W. reports speaker fees from GE. The remaining authors declare no competing interests.
