## [Peer Review File · Communications Biology]

Reviewers' comments:

Reviewer #1 (Remarks to the Author):

Zarkali et al used advanced imaging and analysis approaches to discover brain connectivity changes in PD patients versus controls, and in two PD subgroups, i.e. patients low higher-order visual function (who have a higher risk to develop dementia) and patients with high higher-order visual function. This is a descriptive study which compares a new marker (called "structural-functional coupling") which was derived from different imaging modalities between patients to controls. While this is an interesting observation, it is not completely clear which biological mechanisms are reflected by the investigated marker or whether the marker is relevant for cognition of other functions in PD patients.

The manuscript is well-written and seeks to answer a timely question. The authors use very advanced analyses which were executed with care. The general idea is interesting but the authors rightfully list a series of limitations which, in my eyes, make it difficult to trust the conclusions.

The authors used the term "structure-function coupling" throughout the manuscript. However, I found this term rather misleading because the authors quantified whether a DWI based connectivity metric (extracted from tractography data) correlated with a rs-fMRI based connectivity metric. Thus, "structural connectivity-functional connectivity coupling" is a better term and "DWI connectivity-rsfMRI connectivity coupling" is probably the most correct one.

It would be good to stress that DWI connectivity-rsfMRI connectivity coupling is an abstract macroscopic marker and that it is not clear whether it has a biological meaning or which biological parameters are reflected by this coupling. The authors acknowledge that the DWI connectivity metric (which is based on tractography) is an estimate of monosynaptic connections of the brain which is known to contain many false positives as well as some false negatives. However, this makes a putative biological interpretation of the DWI connectivity-rsfMRI connectivity coupling even more challenging since the DWI connectivity matrix might be subject to a lot of noise. Moreover, it is unclear whether decoupling is causally related to cognition (or other functions) in the PD patients investigated here. In this regard, it is also unclear whether neuromodulatory correlates of structure-function decoupling in PD are really relevant.

This doesn't take away that DWI connectivity-rsfMRI connectivity coupling is a marker which differs (on average) between patients and controls. However, one might wonder whether DWI connectivity-rsfMRI connectivity coupling is more sensitive to differences between PD and controls than each metric on its own.

I found it really difficult to follow how DWI connectivity-rsfMRI connectivity coupling was correlated with cortical expression of neurotransmitter receptor genes. The manuscript would benefit if the authors could extend their description how exactly this data has been analysed. E.g. were all 400 nodes cortical nodes? If not, how many were used for the analysis including gene expression. How were the contrast between PD and controls and the two PD subgroups determined?

Some correlations in figure 5 are significant but, for large datasets, statistical significance is reached even if only a small fraction of the variance is explained. Please discuss whether $\rho < 0.3$ indicates a rather strong or a rather weak effect in the context of the present study (I'm aware that this is not easy since data was ranked).

Minor comments:

Table 1. demographics: Please check the vision chapter because the numbers look wrong (stats might be correct)

Were controls matched to the PD high visual Performers or to the group as a whole?

Reviewer #2 (Remarks to the Author):

Thank you for inviting me to review this manuscript by Zarkali and colleagues, in which the authors analysed the patterns of structural-functional coupling in a cohort of individuals with Parkinson's disease, delineated by either high or low visual function. The authors found differences between PD and controls, where PD has a diminished structure-function coupling across the brain. Additionally, this effect was found to correlate with the anterior-posterior gradient present across the cerebral cortex. Finally, the authors asked whether this effect was correlated with a set of receptors involved in large-scale neuromodulation.

The paper used advanced techniques and up-to date integrative methods to study large-scale patterns effects of PD. However, despite this interesting approach, there were methodological issues within the paper that I feel warrant further consideration.

Introduction

Line 67: The authors may wish to comment on whether the SF coupling in PFC is associated with increase or decrease in executive function.

Line 76: Although I understand that if there is a SF coupling, it was not particularly clear why SF coupling in itself was important or relevant in the pre-frontal cortex in particular, as opposed to structural or functional changes in different regions of the brain.

Line 84-85: In the introduction you mentioned about the anterior-posterior axis of influence of SF coupling, and also through the hierarchical processing gradient, but it was only in the healthy brain, and there was no specific hypothesis with respect to PD.

Line 86: the term 'posterior hypometabolism' was not introduced before this point.

Line 90: Given the focus of the study, I think it's quite important that the authors explain why they chose to interrogate the neuromodulatory system (e.g., they might consider referencing some of the work involving neuromodulatory damage in PD).

Results:

Line 104: it says 33 controls, although in the introduction you mentioned 30 controls.

Line 105-106: The reader would benefit from a better explanation of the tasks used in the study, as it is an important criterion that is used to separate the groups.

Line 107: Along these lines, I think that it's important to clarify the criteria to split the groups, as they are not equal in number.

Also, as they have a different score, it would be important to show the statistics showing the effect

size, as that value would explain the strength of the difference between the two groups.

Line 125: In general, there was insufficient statistical information. For instance, in figure 2C, the authors show three groups, but only the statistic for one comparison it is shown.

Line 135-136: As it is part of the comparison, it is important to show the data for the control group for these regions.

Line 146-147: As in former results, it is important to mention the statistics that were used here.

Line 162-163: it is not clear whether the difference is in the control group or across the population level.

Line 177-186: Perhaps a linear regression conducted in each subject would be offer a more parsimonious test of the data.

Relationship between structure-function decoupling in PD and neurotransmitter receptor gene expression:

Line 187-201: The authors should explicitly state whether the results of the receptor analysis were exploratory (which is my impression) or were hypothesis driven. In addition, recent papers by Zerbi et al (2019) and Richiardi et al (2015) offer demonstrations of how to perhaps better correct for issues inherent to this analysis (such as spatial autocorrelation).

Discussion:

Line 207-208: it is necessary to state that those effects were found when comparing within the PD group, because stated in the way it is, it comes across as if the low visual performers have only a focal effect on the SF coupling across all groups.

Line 226-229: Forgive me if I'm wrong, but in the introduction it was stated that the SF coupling in PFC was an adaptive result and, thus, correlated with higher cognitive abilities. If this is indeed the case, then the explanation offered here is somewhat contradictory.

Response to Reviewers

COMMSBIO-20-2001-T: Organisational and neuromodulatory underpinnings of structure-function decoupling in Parkinson's disease

We thank the editors for considering our manuscript for publication in *Nature Communications Biology*. We were pleased that reviewers found our paper to be “**interesting**” and “**well-written**” and our analyses methods “**advanced**”.

We have now addressed the Reviewers' comments. In particular, we have changed the terminology to structural-functional connectivity coupling (SC-FC coupling) throughout the manuscript, we have provided further methodological details on the two visual experimental tasks and the correlation of regional differences in SC-FC coupling with neurotransmitter receptor gene expression differences. Additionally, we have included further details on SC-FC coupling values in controls and further analysed structural and functional connectivity separately in our cohort for comparison.

We hope that our revisions meet with the Editor's and Reviewers' approval.

Response to Reviewer 1:

Thank you for your thoughtful comments. We were pleased that the Reviewer found our manuscript “**interesting**”, “**well written**” and “**timely**”. We have now addressed your comments below.

Comments:

Comment 1: *“The authors used the term “structure-function coupling” throughout the manuscript. However, I found this term rather misleading because the authors quantified whether a DWI based connectivity metric (extracted from tractography data) correlated with a rs-fMRI based connectivity metric. Thus, “structural connectivity-functional connectivity coupling” is a better term and “DWI connectivity-rsfMRI connectivity coupling” is probably the most correct one.”*

Response: We thank the Reviewer for this comment. We chose the structure-function coupling term for simplicity, in line with a number of other published studies^{1,2}. However, we

agree that this may be confusing to some readers and we have changed this term to structural-functional connectivity coupling or SC-FC coupling throughout the manuscript.

Comment 2: *“It would be good to stress that DWI connectivity-rsfMRI connectivity coupling is an abstract macroscopic marker and that it is not clear whether it has a biological meaning or which biological parameters are reflected by this coupling. The authors acknowledge that the DWI connectivity metric (which is based on tractography) is an estimate of monosynaptic connections of the brain which is known to contain many false positives as well as some false negatives. However, this makes a putative biological interpretation of the DWI connectivity-rsfMRI connectivity coupling even more challenging since the DWI connectivity matrix might be subject to a lot of noise. Moreover, it is unclear whether decoupling is causally related to cognition (or other functions) in the PD patients investigated here. In this regard, it is also unclear whether neuromodulatory correlates of structure-function decoupling in PD are really relevant.*

Response: We agree that both the diffusion weighted imaging and resting state derived connectivity are indirect markers of structural and functional connectivity respectively. To maximise the validity of our results, particularly the structural connectivity estimates derived from diffusion weighted imaging, we have used a strict quality control process, multi-shell data and a higher tensor model which has shown better performance than traditional diffusion tensor imaging metrics³. Nevertheless, to emphasise the Reviewer’s, we have now reiterated more clearly the limitations of using indirect markers of structural and functional connectivity in our revised Discussion (Methodological considerations, Page 11).

Despite the inherent limitations of indirect imaging measures, the regional differences in SC-FC coupling have already been able to provide meaningful insights on the relationship between structure and function in health^{1,4,5} and in disease^{6,7}, and investigating neurotransmitter gene expression profiles in relation to SC-FC coupling has been proposed as a promising area of research in better understanding how the structure-function is modulated⁸.

We have shown widespread SC-FC decoupling in PD with a further significant focal decoupling in PD low visual performers, who are believed to be at highest risk of subsequent dementia^{9,10}. Other studies have also shown a correlation between stronger SC-FC coupling

and improved cognition in health^{1,4}. Taking into account the Reviewer's comments of the relevance of SC-FC coupling in PD, we have now examined how SC-FC coupling in the most affected regions (left and right insula and right calcarine) is related to cognition by examining the relationship of SC-FC coupling in these regions with MOCA scores in patients with PD (spearman correlation coefficient, adjusting for three comparisons made).

We found that MOCA scores in PD were correlated with SC-FC coupling in the right calcarine ($r=0.307$, $p=0.003$, corrected for multiple comparisons $q=0.011$), but not in the insula.

Relationship between SC-FC coupling in the right calcarine gyrus and MOCA scores in patients with Parkinson's Disease (PD). The *p*-value shown is corrected for multiple comparisons (comparison of MOCA scores to 3 regions of interest: right calcarine, left insula and right insula).

These findings have been included in the revised Results section: Widespread structural-functional connectivity decoupling occurs in PD (Page 6) and in Supplementary Figure 3.

Comment 3: *This doesn't take away that DWI connectivity-rsfMRI connectivity coupling is a marker which differs (on average) between patients and controls. However, one might wonder whether DWI connectivity-rsfMRI connectivity coupling*

is more sensitive to differences between PD and controls than each metric on its own.”

Response: We thank the Reviewer for this interesting point. We used SC-FC coupling in this study to shed light on the spatial organisation of structure-function coupling in PD, rather than as a biomarker to distinguish between PD and control or PD low and high visual performers. Assessing the relationship between structural and functional connectivity offers insights into changes in brain organisation that are not revealed by looking at each separately; in particular how hierarchical patterns of organisation in the cortex could contribute to selective vulnerability of brain regions in the presence of degeneration, for example with transmodal and more posterior regions being more vulnerable. Both the structural and functional connectivity changes that occur in PD compared to controls have been investigated by several independent research groups using different complementary methodologies¹⁰⁻¹⁹. We therefore feel that repeating a separate assessment of structural and functional alterations in PD is unlikely to provide any further new insights.

In order to provide the readers with a point of reference for structural and functional connectivity changes in our cohort, however, we have performed network-based statistics to investigate changes in structural and functional connectivity separately. A general linear model was used with contrast of interest including PD versus controls and PD low visual performers versus high visual performers with age and gender included as covariates. Permutation testing with unpaired t-tests was performed with 5000 permutations, calculating a test statistic for each connection. A threshold of $t = 3.1$ as well as family-wise error rate (FWE) of $p < 0.05$ was applied.

We found a subnetwork of reduced functional connectivity strength in PD compared to controls (Figure 2, Supplementary Figure S6), similar to that previously described¹⁵. No significant subnetwork of reduced structural connectivity strength was seen in PD compared to controls. In addition, no significant subnetworks of reduced structural or functional connectivity strength was seen in PD low visual performers compared to PD high visual performers.

We report the results of these additional analyses in Supplementary Figure 6.

Subnetwork of reduced functional connectivity strength in PD compared to controls.

- A. Subnetwork of reduced functional connectivity strength in PD compared to controls ($p=0.003$). The subnetwork was visualised using BrainNetViewer²⁰.*
- B. The nodes of the subnetwork were visualised according to the number of reduced connections; darker red colours represent higher number of connections from that node showing reduced functional connectivity strength (see colour scale).*

Comment 4: “I found it really difficult to follow how DWI connectivity-rsfMRI connectivity coupling was correlated with cortical expression of neurotransmitter receptor genes. The manuscript would benefit if the authors could extend their description how exactly this data has been analysed. E.g. were all 400 nodes cortical nodes? If not, how many were used for the analysis including gene expression. How were the contrast between PD and controls and the two PD subgroups determined?”

Response: Gene expression data for the 31 pre-selected genes encoding neurotransmitter receptors were extracted for each of the 400 cortical nodes of our parcellation. All of the 400 cortical nodes were included in the gene expression analyses. Spearman correlations were performed between the regional differences in SC-FC coupling across the 400 cortical nodes between 1) PD vs controls and 2) PD low vs high visual performers and the gene expression levels for each of the 31 genes, FDR corrected for multiple comparisons. The regional difference in SC-FC coupling between groups was expressed as the vector of difference in SC-FC coupling as visualised in Figure 2B.

We have now clarified the methodology of our gene expression analysis in the Methods: Neurotransmitter receptor gene expression (Pages 15-16) and Statistical comparisons (Page 16).

Comment 5: “Some correlations in figure 5 are significant but, for large datasets, statistical significance is reached even if only a small fraction of the variance is explained. Please discuss whether $\rho < 0.3$ indicates a rather strong or a rather weak effect in the context of the present study (I’m aware that this is not easy since data was ranked).”

Response: We thank the Reviewer for this important point. Regional gene expression changes only account for a moderate fraction of the variance in our dataset, and we have now clarified this in our Results (Table 3, Page 29) and Discussion (Page 10).

Minor Comments:

Comment 1: “Table 1. demographics: Please check the vision chapter because the numbers look wrong (stats might be correct)”

Response: We thank the Reviewer for highlighting a typographical error in Table 1. Contrast sensitivity (Pelli Robson) for PD low visual performers was: mean (std) = 1.7 (0.1). This has now been corrected in Table 1 (Page 27).

Comment 2: “Were controls matched to the PD high visual Performers or to the group as a whole?”

Response: Controls were matched to the PD group as a whole and not to PD high visual performers. We have now clarified this in the Methods: Participants (Page 12).

Response to Reviewer 2:

We thank the Reviewer for their insightful comments. We were pleased that the Reviewer’s found our approach “**interesting**” and “**advanced**”. We have addressed the Reviewer’s specific comments below.

Introduction:

Comment 1: *“Line 67: The authors may wish to comment on whether the SF coupling in PFC is associated with increase or decrease in executive function.”*

Response: We thank the Reviewer for this comment. Increased SC-FC coupling in the prefrontal cortex was associated with increase in executive function. We have now added this to the Introduction, Page 3.

Comment 2: *“Line 76: Although I understand that if there is a SF coupling, it was not particularly clear why SF coupling in itself was important or relevant in the pre-frontal cortex in particular, as opposed to structural or functional changes in different regions of the brain.”*

Response: The studies we cite had not shown a directional relationship from frontal to posterior brain regions, but rather that connections between these regions are affected in PD with cognitive impairment. We have now amended the manuscript to reflect this point (Introduction, Page 3). We have also clarified that SC-FC decoupling along this posterior – anterior (and also unimodal – transmodal) axes were of greatest interest, rather than decoupling in any one particular region (Introduction, Page 4).

Comment 3: *“Line 84-85: In the introduction you mentioned about the anterior-posterior axis of influence of SF coupling, and also through the hierarchical processing gradient, but it was only in the healthy brain, and there was no specific hypothesis with respect to PD”.*

Response: We hypothesised that in PD SC-FC decoupling would occur along one of two hypothesised directions: 1) across the unimodal-transmodal hierarchical gradient of SC-FC decoupling that is seen in health with transmodal regions becoming even more decoupled in PD^{1,4,5,21}; or 2) along the anterior-to-posterior (A-P) axis with decoupling more prominent in posterior regions. This hypothesis was based on the posterior distribution of metabolic and connectivity changes seen in PD and specifically linked to dementia in PD.^{11,22–26} We have now restructured our Introduction to clarify this (Page 4).

Comment 4: *“Line 86: the term ‘posterior hypometabolism’ was not introduced before this point.”*

Response: Thank you for bringing this to our attention. We have now introduced the concept of posterior hypometabolism earlier in the introduction and specifically linked it with

Parkinson's dementia (Introduction, Page 3). We hope that in the section describing our hypotheses, the basis for expecting a change along the anterior-to-posterior axis is now clearer (Page 4).

Comment 5: *“Line 90: Given the focus of the study, I think it's quite important that the authors explain why they chose to interrogate the neuromodulatory system (e.g., they might consider referencing some of the work involving neuromodulatory damage in PD).”*

Response: Although PD is classically associated with altered dopaminergic transmission, there is considerable neurochemical evidence to implicate other neurotransmitter systems particularly in relation to cognitive impairment. Clarifying the affected neurotransmitter systems in this patient group who are at highest risk of subsequent cognitive decline could provide further support for investigating the benefit of therapies targeting not only the dopaminergic and cholinergic system (currently the mainstay of treatment) but also the serotonergic and noradrenergic system in PD dementia. We now include more details on the rationale for this analysis in our Introduction (Page 4).

Results:

Comment 1: *“Line 104: it says 33 controls, although in the introduction you mentioned 30 controls.”*

Response: We thank the Reviewer for highlighting this typographical error; indeed 30 controls were included and this has now been corrected (Page 4).

Comment 2: *“Line 105-106: The reader would benefit from a better explanation of the tasks used in the study, as it is an important criterion that is used to separate the groups.”*

Response: We had not included these details due to word count constraints and as the tasks had been previously described^{9,10,13,27,28}. However, we agree with the Reviewer that more information is needed to help understand the tasks that were used to define the groups. We have now provided more details to the methods describing the visual tasks and the rationale for choosing these particular measures. (Methods, Pages 12-13). We also include examples of the stimuli used in Supplementary Figure S1.

Comment 3: “Line 107: Along these lines, I think that it’s important to clarify the criteria to split the groups, as they are not equal in number.

Also, as they have a different score, it would be important to show the statistics showing the effect size, as that value would explain the strength of the difference between the two groups.”

Response: We defined the poor visual performers as those patients who performed worse than the group median score for both tasks. This allowed us to capture the group of patients who performed consistently poorly in the higher order visual tasks. As they had to perform worse than the median in both tasks, this gave us an uneven split, with fewer patients in the

poor vision group. We have now clarified this in the revised methods section (Pages 12-13). Group performance details for participants with Parkinson’s disease are seen below, including z-scored performance (and now provided as Supplementary figure S2). There was no significant difference between the whole of patients with PD and controls in either the *Cats and Dogs* ($\rho=0.168$, $p=0.057$) or the *Biological motion* task ($\rho=0.025$, $p=0.778$).

Performance of patients with Parkinson’s disease in the visual tasks.

A. Distribution of performance in the Cats and Dogs task. Orange line highlights the median performance in this task.

B. Distribution of performance in the Biological motion task. Orange line highlights the median performance in this task.

C. Distribution of the combined Z score of performance in the Biological motion and Cats and Dogs task ($Z_{score} = Z_{Biolmotion} + Z_{cats\&dogs}$). The distribution on the left of the red line

represents the patients who performed worse than the median (left of the orange lines in A and B) for both the Cats and Dogs and the Biological motion task. These were classified as PD low visual performers.

Following the Reviewer's comments, we have now included these additional details on group performance on these tasks in our Supplementary Material (Supplementary Figure S2).

Comment 4: *“Line 125: In general, there was insufficient statistical information. For instance, in figure 2C, the authors show three groups, but only the statistic for one comparison it is shown.”*

Response: The results of the statistical comparisons shown in Figure 2C were presented fully in the Results, but had been omitted from Figure 2C's Legend. Specifically, we describe widespread structural-functional connectivity decoupling in PD (in the first paragraph for PD vs controls and third paragraph for PD low vs PD high visual performers). PD low visual performers showed reduced global SC-FC coupling than controls (mean 0.469 vs 0.544, $p=0.002$); PD high visual performers showed reduced global SC-FC coupling than controls (mean 0.492 vs 0.544, $p=0.005$); there was no significant difference between PD low and high visual performers (mean 0.469 vs 0.492, $p=0.415$). Following the Reviewer's comment, we now further include this statistical information in the legend of Figure 2C.

Comment 5: *“Line 135-136: As it is part of the comparison, it is important to show the data for the control group for these regions.”*

Response: We agree that showing the data for the control group is important. To facilitate comparison with the changes seen in PD, the mean SC-FC coupling in control participants had been visualised in Figure 2A but this was not explicit in the Figure title. We have now clarified this oversight, emphasising that the figure shows both controls and PD. In addition, taking into account the Reviewer's comment we have also included the mean SC-FC coupling values for our comparisons of interest (PD vs controls and PD low vs PD high visual performers) for these regions in Table 2 (Page 30).

Comment 6: *“Line 146-147: As in former results, it is important to mention the statistics that were used here.”*

Response: We used diffusion map embedding to derive cortical gradients from structural and functional connectivity matrices. After normalisation of the similarity matrix, diffusion map

embedding computes the first ten components with the largest eigenvalues and the accompanying eigenvectors (gradients) of the diffusion operator²⁹. For each gradient the explained variance can be directly derived from its eigenvalue by dividing the gradient's eigenvalue by the sum of all eigenvalues²⁹.

We now specify the method for deriving explained variance from gradients in our Methods: Gradient analysis (Page 15).

Comment 7: *“Line 162-163: it is not clear whether the difference is in the control group or across the population level.”*

Response: We apologise that this was not clear. This section examined the spatial organisation of structural and functional connectivity in the first two gradients in healthy controls, to confirm the underlying normal gradients of macroscale cortical organisation in health and to test whether these gradients contribute to selective vulnerability of SC-FC decoupling in PD. We have now made this clearer in the Results (Page 6).

Comment 8: *“Line 177-186: Perhaps a linear regression conducted in each subject would be offer a more parsimonious test of the data.”*

Response: Here, we aimed to clarify the spatial organisation of the first and second principal gradients in the healthy human brain, specifically the alignment of the first principal gradient with the anterior-posterior (A-P) axis and the second principal gradients with the unimodal-transmodal axis. We then correlated the mean gradient values in controls with individual participants SC-FC coupling values across cortical regions. To derive gradients of spatial organisation in health, we only assessed our control participants (n=30) at group level by comparing the mean first gradient value and A-P axis coordinate for each of the 400 cortical nodes and the mean second gradient value with the unimodal-transmodal hierarchy level for each of the 400 cortical nodes. We feel that a group-level analysis of our controls as a whole is a more accurate representation of the “healthy adult gradient”. However, to ensure that this was not driven by a few individuals, we have now performed the same comparisons at an individual level for each of the 30 control participants. The range of these correlations (Spearman's rho) at individual level is seen in the table below (in all the 30 controls, p-spin <0.005).

Cortical gradient alignment at individual level (range of Spearman's rho).		
	First principal gradients	
	Structural	Functional
A-P axis	-0.651, -0.572	-0.684, -0.267
	Second principal gradients	
	Structural	Functional
Unimodal-Transmodal axis	0.372, 0.518	0.239, 0.749

In the revised manuscript, we have now reported the range of individual correlations in our control participants in the Results: Defining structural and functional gradients of macroscale cortical organisation in health (Pages 6-7).

Comment 9: *“Relationship between structure-function decoupling in PD and neurotransmitter receptor gene expression: Line 187-201: The authors should explicitly state whether the results of the receptor analysis were exploratory (which is my impression) or were hypothesis driven. In addition, recent papers by Zerbi et al (2019) and Richiardi et al (2015) offer demonstrations of how to perhaps better correct for issues inherent to this analysis (such as spatial autocorrelation).”*

Response: Linking SC-FC decoupling with neurotransmitter receptor gene expression was an exploratory, data driven analysis. We now specify this in the Introduction (Page 4).

We agree that the point of spatial autocorrelation is an important one, that has emerged particularly with recent observations³⁰. A specific concern, however, is that there is a risk of false positive findings arising spuriously when linking gene expression and brain imaging data, due to gene-gene co-expression and spatial autocorrelation whereby neighbouring brain regions show higher similarity of gene expression. To mitigate against this possibility of false positive bias and to ensure that the relationship between regional difference in SC-FC coupling and neurotransmitter gene expression is higher than what would be expected by chance, we have now carried out additional spatial permutation testing. Specifically we performed 1000 permutations of random spatial permutations³¹ of our regional SC-FC coupling differences in PD versus controls and PD low versus PD high visual performers. We calculated Spearman correlation coefficient between extracted regional SC-FC values and gene expression levels of the 31 selected neurotransmitter receptor genes. The permutation-based p-value (pspin) was calculated as the proportion of times that the null correlation

coefficients were greater than the empirical coefficients³¹. In all cases (all 31 receptor genes, for both PD vs controls and PD low vs PD high visual performers) gene expression levels were correlated with SC-FC values significantly more than expected by chance (p-spin <0.001), leading to no change in the receptor results and confirming the validity of our findings. We now report this in Methods (Statistical comparisons, Page 17) and Table 3 (Page 31).

Discussion:

Comment 1: *“Line 207-208: it is necessary to state that those effects were found when comparing within the PD group, because stated in the way it is, it comes across as if the low visual performers have only a focal effect on the SF coupling across all groups.”*

Response: We now specify that focal SC-FC decoupling refers to PD low versus PD high visual performers (Discussion: Page 8).

Comment 2: *“Line 226-229: Forgive me if I’m wrong, but in the introduction it was stated that the SF coupling in PFC was an adaptive result and, thus, correlated with higher cognitive abilities. If this is indeed the case, then the explanation offered here is somewhat contradictory.”*

Response: Current models suggest that a weaker SC-FC coupling in transmodal regions may be an adaptive response allowing these regions to serve more varied functions and hence more adaptive and flexible cognition³². Even though this may be the case in health, in the presence of disease, this weaker coupling may make these higher-order, transmodal regions even less constrained by sensory processing resulting in the complex and often positive phenomena accompanying PD dementia, including visual hallucinations and delusions. We now clarify this in the Discussion (Page 9).

References:

1. Baum, G. L. *et al.* Development of structure–function coupling in human brain networks during youth. *Proc. Natl. Acad. Sci. U. S. A.* **117**, 771–778 (2020).
2. Preti, M. G. & Van De Ville, D. Decoupling of brain function from structure reveals regional behavioral specialization in humans. *Nat. Commun.* **10**, (2019).
3. Delettre, C. *et al.* Comparison between diffusion MRI tractography and histological tract-tracing of cortico-cortical structural connectivity in the ferret brain. *Netw. Neurosci.* **3**, 1038–1050 (2019).
4. Vázquez-Rodríguez, B. *et al.* Gradients of structure–function tethering across neocortex. *Proc. Natl. Acad. Sci.* **116**, 21219–21227 (2019).
5. Preti, M. G. & Van De Ville, D. Decoupling of brain function from structure reveals regional behavioral specialization in humans. *Nat. Commun.* **10**, (2019).
6. Koubiyr, I. *et al.* Dynamic modular-level alterations of structural-functional coupling in clinically isolated syndrome. *Brain* **142**, 3428–3439 (2019).
7. McColgan, P. *et al.* White matter predicts functional connectivity in premanifest Huntington’s disease. *Ann. Clin. Transl. Neurol.* **4**, 106–118 (2017).
8. Suárez, L. E., Markello, R. D., Betzel, R. F. & Misic, B. Linking Structure and Function in Macroscale Brain Networks. *Trends Cogn. Sci.* **24**, 302–315 (2020).
9. Leyland, L.-A. *et al.* Visual tests predict dementia risk in Parkinson disease. *Neurol. Clin. Pract.* 10.1212/CPJ.0000000000000719 (2019) doi:10.1212/CPJ.0000000000000719.
10. Weil, R. S. *et al.* Neural correlates of early cognitive dysfunction in Parkinson’s disease. *Ann. Clin. Transl. Neurol.* **0**, (2019).
11. Kamagata, K. *et al.* Connectome analysis with diffusion MRI in idiopathic Parkinson’s disease: Evaluation using multi-shell, multi-tissue, constrained spherical deconvolution. *NeuroImage Clin.* **17**, 518–529 (2018).

12. Kawabata, K. *et al.* Distinct manifestation of cognitive deficits associate with different resting-state network disruptions in non-demented patients with Parkinson's disease. *J. Neurol.* **265**, 688–700 (2018).
13. Zarkali, A. *et al.* Fiber-specific white matter reductions in Parkinson hallucinations and visual dysfunction. *Neurology* (2020) doi:10.1212/WNL.0000000000009014.
14. Rau, Y.-A. *et al.* A longitudinal fixel-based analysis of white matter alterations in patients with Parkinson's disease. *NeuroImage Clin.* **24**, (2019).
15. Abós, A. *et al.* Discriminating cognitive status in Parkinson's disease through functional connectomics and machine learning. *Sci. Rep.* **7**, 45347 (2017).
16. Bledsoe, I. O., Stebbins, G. T., Merkitch, D. & Goldman, J. G. White matter abnormalities in the corpus callosum with cognitive impairment in Parkinson disease. *Neurology* **91**, e2244–e2255 (2018).
17. Deng, B. *et al.* Diffusion tensor imaging reveals white matter changes associated with cognitive status in patients with Parkinson's disease. *Am. J. Alzheimers Dis. Other Demen.* **28**, 154–164 (2013).
18. Hanganu, A. *et al.* White matter degeneration profile in the cognitive cortico-subcortical tracts in Parkinson's disease. *Mov. Disord. Off. J. Mov. Disord. Soc.* **33**, 1139–1150 (2018).
19. Melzer, T. R. *et al.* White matter microstructure deteriorates across cognitive stages in Parkinson disease. *Neurology* **80**, 1841–1849 (2013).
20. Xia, M., Wang, J. & He, Y. BrainNet Viewer: A Network Visualization Tool for Human Brain Connectomics. *PLOS ONE* **8**, e68910 (2013).
21. Paquola, C. *et al.* Microstructural and functional gradients are increasingly dissociated in transmodal cortices. *PLOS Biol.* **17**, e3000284 (2019).

22. Zarkali, A. *et al.* Fibre-specific white matter reductions in Parkinson's hallucinations and visual dysfunction. *Neurology* (**in press**), (2020).
23. Garcia-Garcia, D. *et al.* Posterior parietooccipital hypometabolism may differentiate mild cognitive impairment from dementia in Parkinson's disease. *Eur. J. Nucl. Med. Mol. Imaging* **39**, 1767–1777 (2012).
24. González-Redondo, R. *et al.* Grey matter hypometabolism and atrophy in Parkinson's disease with cognitive impairment: a two-step process. *Brain* **137**, 2356–2367 (2014).
25. Tang, Y. *et al.* Cerebral Metabolic Differences Associated with Cognitive Impairment in Parkinson's Disease. *PloS One* **11**, e0152716 (2016).
26. Bohnen, N. I. *et al.* Cerebral glucose metabolic features of Parkinson disease and incident dementia: longitudinal study. *J. Nucl. Med. Off. Publ. Soc. Nucl. Med.* **52**, 848–855 (2011).
27. Weil, R. S. *et al.* The Cats-and-Dogs test: A tool to identify visuoperceptual deficits in Parkinson's disease. *Mov. Disord.* **32**, 1789–1790 (2017).
28. Weil, R. S. *et al.* Assessing cognitive dysfunction in Parkinson's disease: An online tool to detect visuo-perceptual deficits. *Mov. Disord.* **33**, 544–553 (2018).
29. Vos de Wael, R. *et al.* BrainSpace: a toolbox for the analysis of macroscale gradients in neuroimaging and connectomics datasets. *Commun. Biol.* **3**, 1–10 (2020).
30. Fulcher, B. D., Arnatkevičiūtė, A. & Fornito, A. Overcoming bias in gene-set enrichment analyses of brain-wide transcriptomic data. *bioRxiv* 2020.04.24.058958 (2020)
doi:10.1101/2020.04.24.058958.
31. Alexander-Bloch, A. *et al.* On testing for spatial correspondence between maps of human brain structure and function. *NeuroImage* **178**, 540–551 (2018).

32. Wendelken, C., Ferrer, E., Whitaker, K. J. & Bunge, S. A. Fronto-Parietal Network Reconfiguration Supports the Development of Reasoning Ability. *Cereb. Cortex N. Y. N* 1991 **26**, 2178–2190 (2016).

REVIEWERS' COMMENTS:

Reviewer #1 (Remarks to the Author):

The authors provided a very thorough revision and addressed virtually all comments. This is a sound study within the principle limitations of the methods available in human patients. The authors discuss these limitations in the "Methodological considerations and future directions" paragraph which indicates that there are still many open questions.

Reviewer #2 (Remarks to the Author):

The authors have adequately addressed my concerns.